# Water Vapor Calibration: Using a Raman Lidar and Radiosoundings to Obtain Highly Resolved Water Vapor Profiles

**Birte Solveig Kulla [1,2,*]** and **Christoph Ritter [1,*]**

[1]   Alfred-Wegener-Institut, Telegrafenberg A45, 14473 Potsdam, Germany
[2]   Institut für Geophysik und Meteorologie, Universität zu Köln, Pohligstr. 3, 50969 Cologne, Germany
*   Correspondence: birte.kulla@awi.de (B.S.K.); christoph.ritter@awi.de (C.R.)

**Abstract:** We revised the calibration of a water vapor Raman lidar by co-located radiosoundings for a site in the high European Arctic. For this purpose, we defined robust criteria for a valid calibration. One of these criteria is the logarithm of the water vapor mixing ratio between the sonde and the lidar. With an error analysis, we showed that for our site correlations smaller than 0.95 could be explained neither by noise in the lidar nor by wrong assumptions concerning the aerosol or Rayleigh extinction. However, highly variable correlation coefficients between sonde and consecutive lidar profiles were found, suggesting that small scale variability of the humidity was our largest source of error. Therefore, not all co-located radiosoundings are useful for lidar calibration. As we assumed these changes to be non-systematic, averaging over several independent measurements increased the calibration's quality. The calibration of the water vapor measurements from the lidar for individual profiles varied by less than $\pm 5\%$. The seasonal median, used for calibration in this study, was stable and reliable (confidence $\pm 1\%$ for the season with most calibration profiles). Thus, the water vapor mixing ratio profiles from the Koldewey Aerosol Raman Lidar (KARL) are very accurate. They show high temporal variability up to 4 km altitude and, therefore, provide additional, independent information to the radiosonde.

**Keywords:** lidar; Raman shift; AWIPEV; Svalbard; water vapor; water vapor calibration; radiosonde; GRUAN; Vaisala; atmosphere

## 1. Introduction

In the Arctic, average temperatures rise twice as fast as on global average; the so-called "Arctic Amplification" of global warming. This regional pattern of global warming with its pronounced warming north of the Arctic Circle is associated with various feedbacks and still subject of ongoing research [1]. One of the hot spots of warming is the eastern European Arctic between Spitsbergen and the Kara Sea. In Ny-Ålesund at the west coast of Spitsbergen (79.9°N and 11.9°E), where the measurements for this study took place, a warming of 3 K per decade in winter (DJF) was observed over the past 26 years [2]. Roughly a quarter of this warming can be attributed to an increased advection of warm and moist air from the North Atlantic into the European Arctic [3].

Water vapor is the strongest and most variable greenhouse gas [4] and its vertical distribution determines its radiative impact [5]. A clear impact (at least of stratospheric) water vapor on the climate of the last decades has been proven [6]. Assessing the role of tropospheric water vapor, however, is very challenging as feedbacks with clouds and aerosol must be considered and are not fully understood yet. A systematic deviation between aerosol extinction from remote sensing and in-situ measurements in the Arctic boundary layer [7] indicates that hygroscopic growth of aerosol needs to be investigated

over the full atmospheric column. Hence, monitoring water vapor, especially in the Arctic atmosphere, is an important task. Addressing the integrated water vapor under the usually very dry conditions of the Arctic atmosphere in winter is not easy and subject of ongoing research [8].

There are two major approaches to measure water vapor with lidar systems: either using the differential absorption technique (e.g., [9]) or, as was done in this study, using the Raman scattering effect (e.g., [10–12]). Calculating the ratio between the inelastically scattered lidar profile of water vapor and nitrogen, with a minor correction for wavelength-dependent extinction, gives a signal proportional to the water vapor mixing ratio (e.g., [13]). To derive the humidity from the lidar signal, a calibration is necessary to link the ratio of the lidar profiles to the actual water vapor mixing ratio.

This calibration can be done by either measuring the transmission of the different detection branches of the lidar or comparison of the lidar product with external instruments. For the first method, determination of the transmission of both branches (nitrogen and water vapor) of the lidar system [14] has been proposed. For this transmission measurement, skylight or a precise lamp [15] may be used. While this method is completely independent of other instruments, it introduces more components to the lidar and more effort during data acquisition. The second method uses humidity profiles from external instrumentation such as radiosondes (e.g., [16,17]), microwave (e.g., [18,19] or photometer data (e.g., [20] for calibration. In addition, a hybrid technique, employing both a tungsten lamp and radiosonde, has also been introduced [21].

While principally a suite of many different instruments measure the water vapor at the supersite of Ny-Ålesund [22], not all of them are appropriate for lidar calibration. GPS measurements are handicapped by the large footprint of satellites in comparison to KARL at our coastal site with challenging orography posing problems [23].

A HATPRO microwave radiometer only covered a short period of parallel measurements to the lidar. In addition, radiometer humidity profiles have high uncertainties [24] and the integrated water vapor, which is considered to be more reliable, cannot be used for calibration because KARL lacks exact measurement in the lowermost 350 m. Thus, with the incomplete overlap of the telescope, a reliable integrated signal cannot be derived.

In addition, (star-)photometer measurements of water vapor in the polar night have not been analyzed in detail yet and can therefore not be used for lidar calibration. Thus, we chose radiosounding measurements for lidar calibration in this work as they provide continuous height-resolved data at high accuracy. Its in-situ measurements have the huge advantage of high vertical resolution, which can then be resampled to the chosen resolution of our lidar. Disadvantages and uncertainties of radiosoundings are essentially eliminated by carefully choosing calibration constraints, which are explained in this work. For Ny-Ålesund, a calibration technique has been established with data from the dark seasons 2015/2016 and 2016/2017. Only few and short measurements with simultaneous radiosounding measurements were available then. However, in 2018, parallel to the Year Of Polar Prediction (YOPP) campaign with a higher frequency of radiosoundings, longer lidar measurements were carried out. Thus, in winter 2018, 451 hours of lidar measurement were acquired with 21 simultaneous profiles of radiosoundings, which were according to our specifications suited for lidar calibration.

Hence, the aim of this work was to revise the water vapor calibration of a lidar by radiosondes for the Arctic site of Ny-Ålesund. The precision with which a lidar calibration can be performed was determined for our system. After a quick overview of the already established theory, we explain our calibration constraints, compare the GRUAN to the Vaisala humidity and present an error analysis for the mixing ratio profile in the lidar. By using a correlation coefficient between the mixing ratio profiles of radiosonde and lidar, it is shown that correlations below 0.95 could not be explained by noise in the data but indicate different meteorological conditions even for contemporaneous profiles.

## 2. Theory and Background

The Koldewey Aerosol Raman lidar (KARL) is a multi-wavelength Raman lidar for tropospheric and stratospheric research. Among others, the system is part of the Network for the Detection of

Atmospheric Composition Change (NDACC, see: http://www.ndsc.ncep.noaa.gov/), to which it contributes with aerosol properties. The laser emits laser pulses at 1064 nm (IR), 532 nm (VIS) and 355 nm (UV) at 50 Hz with approximately 10 W per wavelength. Raman shifted lines are detected at 407 nm and 660 nm as well as at 387 nm and 607 nm, for the Stokes lines from $H_2O$ and $N_2$ molecules, respectively. The field-of-view of the 70 cm collecting telescope is about 1 mrad (for more details to the setup, see [25]). With the dry atmospheric conditions during polar night, only the counting signal is used and channel combination problems, as discussed in [26], do not have to be taken into account. As the UV signal is less noise-prone, the focus of this work lies on the signal from the 407 nm and 387 nm channels. The transmission-corrected signals from those two channels are proportional to the water vapor and nitrogen density ($\rho H_2O$ and $\rho N_2$) in the respective height bin. As nitrogen is homogeneously distributed in the atmosphere, it is also proportional to the density of dry air. Therefore, the ratio between the two densities is proportional to the volume mixing ratio of water vapor (VMR) [27], which is then also proportional to the mass mixing ratio (MMR).

$$VMR = \frac{\rho_{H_2O}}{\rho_{air}} = \frac{\rho_{H_2O}}{K_{N_2 \to air} \, \rho_{N_2}} = C \, \frac{T^{Ray}_{407} \cdot T^{aer}_{407} \cdot P_{407}}{T^{Ray}_{387} \cdot T^{aer}_{387} \cdot P_{387}} \equiv C \cdot S_{lidar} \tag{1}$$

Here, $C$ is the lidar calibration constant that needs to be fixed by comparison with additional measurements and $S_{lidar}$ is the uncalibrated water vapor signal from the lidar. The expressions $T^{xx}_{387}$ and $T^{xx}_{407}$ denote the transmission of the backscattered light through the atmosphere due to Rayleigh and aerosol extinction at the two wavelengths ($\lambda$). The aerosol transmission depends on the volumetric aerosol extinction coefficient $\alpha^{aer}_{\lambda}$ by:

$$T_\lambda = exp\left(-\int_{z_0}^{z} \alpha^{aer}_\lambda \, d\hat{z}\right) > exp\left(-AOD_\lambda\right) \tag{2}$$

As lidar systems usually do not sound the entire atmosphere but only a limited vertical range, from $z_0$ to $z$ in Equation (2), the integral of the aerosol extinction is always smaller than the aerosol optical depth (AOD) measured by photometers.

Frequently, the aerosol extinction follows a power-law of wavelength with the so-called Ångström exponent A with A < 0. Hence, the ratio of the aerosol transmission terms at the two wavelengths can be expressed by:

$$\frac{T^{aer}_{407}(z)}{T^{aer}_{387}(z)} = exp\left(\int_{z_0}^{z} \alpha^{aer}_{387} \cdot \left(1 - \left(\frac{407}{387}\right)^A\right) d\hat{z}\right) \tag{3}$$

Thus, the error in the water vapor retrieval due to aerosol rises with the aerosol load ($\alpha^{aer}_{387}$) and its spectral slope (A). A large extinction caused by small particles has the higher impact, whereas sub-visible clouds (small extinction, A $\approx$ 0) are not critical. In our work, we set A = $-1.2$, which is close to the long-term average for Ny-Ålesund in March [28].

Given the low water vapor content in the atmosphere above Ny-Ålesund, the signal is very weak and measurements are limited to the dark period. For this study, we found measurements with a sun elevation below $-10°$ suitable. To assess data quality, the signal-to-noise ratio (SNR) was calculated. Therefore, we expressed the lidar signal $P$ in the units of photons, which are clearly identified as individual "steps" in the counting signal. Let the constant $c$ be the conversion between one photon and the measured lidar signal $P$ in arbitrary units. Then, $c \cdot P$ is the lidar signal in the unit photons. (For Licel transient recorders, c might be the inverse of laser shots written in one data file.) In these units, the error of the lidar signal due to photon noise can be expressed by the standard deviation, which is the square root of the signal $c \cdot P$. Moreover, we considered an electronic, or photon-independent error source, which here is called $E$ [29]. Apart from all photon-independent noise sources, $E$ considers effects of a wrong background subtraction as well. The SNR then is defined as

$$SNR = \frac{cP}{\sqrt{cP} + cE} \tag{4}$$

Data for this work were processed with 60 m / 10 min resolution without any further filtering during the evaluation.

## 3. Calibration

As the water vapor signal (P407) is weaker than the nitrogen signal (P987) by much more than one order of magnitude, only the quality of the water vapor signal (P407) was assessed for further analysis. The main source of noise is sunlight. Here, for the resolution of 60 m and 10 min, we mostly found data with sun altitudes below $-10°$ suitable. Depending on the signal strength, the signal-to-noise ratio SNR typically dropped below 10 at an altitude between 3000 and 4000 m. At the expense of resolution, higher altitudes can be reached and profiles during twilight might fulfill the criteria for calibration. However, due to the observed high variability of the humidity, this is only recommendable to some extent.

In addition to this noise in the lidar signal, possible error sources for the calibration are: incorrect measurements by the radiosonde, inaccuracies when converting relative humidity (rh) to the water vapor mixing ratio, and differences in probed air masses, either due to the drift of the radiosonde or due to temporal differences. To minimize the effects these errors might have on the calibration, five criteria were established. Table 1 gives an overview on the requirements made to the data for them to be considered suitable for calibration.

**Table 1.** Requirements for data to be elected for calibration.

| Requirement | Restriction |
|---|---|
| no/little noise | SNR of P407 > 10 |
| not within the incomplete lidar overlap | altitude > 400 m |
| not within a range of known radiosounding bias | not within a cloud (rh < 0.9) and no dry bias (T > −40 °C) |
| not too much time difference | using the measurement closest to radiosounding start time, max. 1 h time difference |
| both measurements have probed similar air masses | Pearson Correlation Coefficient of log > 0.95 (min. 20 pairs per profile have to fulfill the criteria) |

Known biases in radio sounding measurements [30] were excluded by using measurements with temperatures above $-40$ °C and relative humidity below dew point. Moreover, only very reliable lidar measurements at heights with a complete overlap and SNR above 10 are taken into account. We show in Section 6 that this condition guarantees for our data that the remaining noise in the lidar signal does not degrade the retrieval of the calibration constant further.

Furthermore, a Pearson correlation between the logarithm of the two mixing ratio profiles (radio sounding and lidar) was calculated to assure that the same or at least very similar air masses were probed. This is relevant to correct for the sonde's drift and possible small-scale fluctuations of the humidity. The correlation between the signal ratio ($S_{\text{lidar}}$) and the water vapor mixing ratio from the radiosonde ($w$) is known to be linear [31], thus the correlation between $S_{\text{lidar}}$ and $w$ should be unity if noise is not an issue and both measurements have indeed probed the same air masses. As very low values have an equally high impact on the calibration as high values, logarithmic profiles are compared. If the correlation was higher than 0.95, the profiles were considered to match and the air masses had hardly changed between the two measurements. The limit of r > 0.95 is justified in Section 6 by an error estimation.

A example of a consistent mixing ratio profile between lidar and radiosonde is depicted in Figure 1. The heights in which the different rules would be applied are indicated. A valid determination of the mixing ratio is possible for SNR lower than 10. The above-mentioned criteria to exclude data points for calibration are meant to be strict. Only if all of them were given, the corresponding data point was used to determine the lidar calibration constant.

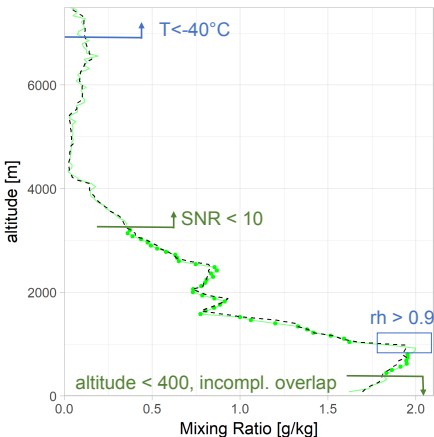

**Figure 1.** Exemplary lidar profile used for calibration (20 February 2018). Bins used for calibration ($z_i$) are highlighted with green points. Conditions under which heights are excluded from calibration as in Table 1 due to deficiencies in lidar (green) or radiosonde data (blue) are marked.

Figure 2 shows this Pearson correlation coefficient between the logarithms of $w_{RS}$ and $S_{\mathrm{lidar}}$. During a period of consistently clear sky in March 2018, lidar observations were compared to the radiosonde closest in time. Correlations very close to unity occurred when measurements were very close in time and air masses were relatively constant (see Figure 7 for comparison). With increasing temporal difference between the measurements, the correlation decreased, sometimes very rapidly. The profiles closest to the start time of the radiosonde were considered to be the best fit. There were also occasions where co-located lidar and radiosonde measurements did not show a good agreement in the mixing ratio profile. In this case, profiles were excluded from the calibration. We come back to this fact in Section 6 and show that small scale fluctuations of the humidity profile are the most probable reason for this, as neither noise in the lidar nor the treatment of aerosol or the Rayleigh atmosphere could explain this variance.

It can be also seen in Figure 2 that sometimes the correlation oscillated around 0.95 for our data. For this reason, we included the additional criterion of a maximum time difference of one hour between lidar and sonde for calibration to guarantee that the same air masses were probed.

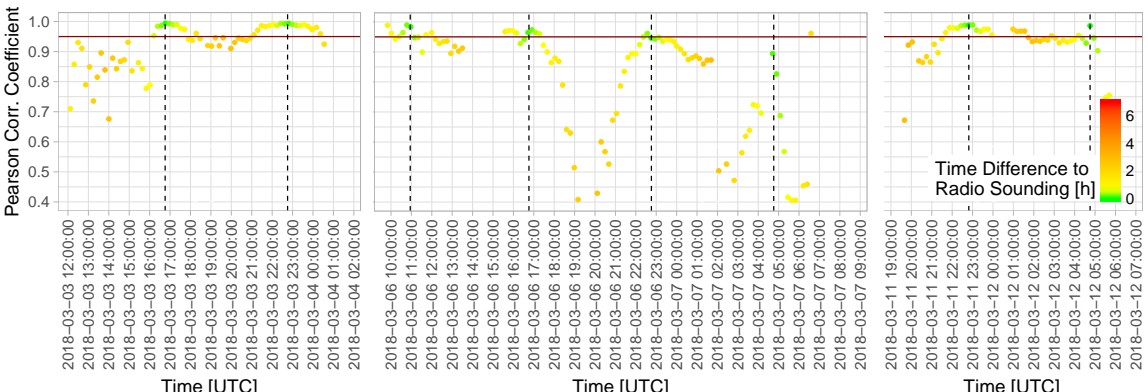

**Figure 2.** Pearson correlation coefficient between $\log(w_{RS})$ and $\log(S_{\mathrm{lidar}})$ at heights where data would fulfill all other criteria to be selected for calibration ($z_i$) during the nights from 3 March 2018 to 4 March 2018, 6 March 2018 to 7 March 2018 and 11 March 2018 to 12 March 2018. Colors indicate the temporal difference between the start of the radiosonde and the lidar measurement. Red line shows the threshold for minimal correlation applied here. Start of radiosounding measurement indicated by black, dashed line.

For all remaining data-pairs $(w_{RS}(t_i, z_i), S_{\mathrm{lidar}}(t_i, z_i))$, a possible correction factor $C_i$ was calculated.

$$w_{RS}(t_i, z_i) = S_{\mathrm{lidar}}(t_i, z_i) C_i$$
$$C_i = \frac{w_{RS}}{S_{\mathrm{lidar}}} \tag{5}$$

$C_i$ varies according to the deviations discussed above (see also Figure 4), and the median is relatively stable. Using the median of all those correction factors ($C_i$) that were needed at the particular heights $z_i$ and fulfill the criteria specified in Table 1 can give one correction factor per profile $t_i$ (dark green I in Figure 3).

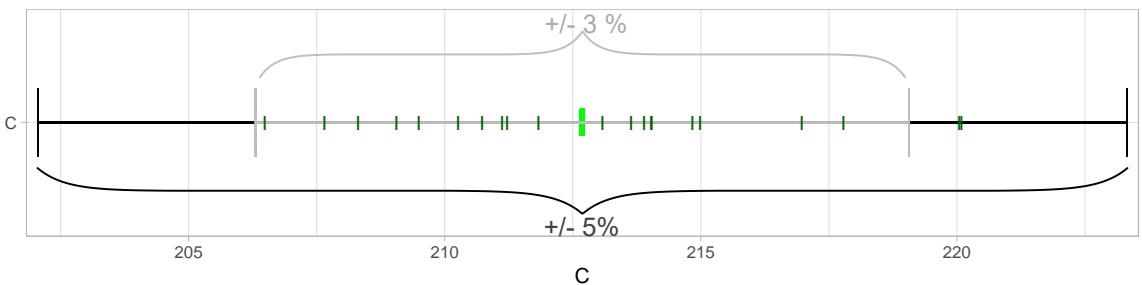

**Figure 3.** Calibration factor for winter 2017/2018. Light green I: median of all $C_i$; dark green I: median of $C_i$, when using only one profile at a time; grey: seasonal median of $C_i$ ±3%; black: seasonal median of $C_i$ ±5%.

This daily $C$ varied within an order of ±3% and was always below ±5%. As we assumed the sources of error to be non-systematic, we could increase the quality of the calibration by increasing the sample size. We found no seasonal drift in these profile specific calibration factors (dark green), and, thus, it was valid to calculate the median over all correction factors at the specified heights in all suitable profiles ($\widetilde{C_i}$, light green I in Figure 3) to determine one calibration factor per season.

We assumed that, due to consistent calibration criteria, all daily calibration constants were equal and that all errors were non-systematic (such as noise in the lidar data and drift of the sonde, which over a season may with equal frequency lead to a dry or wet bias in the sonde). If the lidar was at constant settings over time, the error of a seasonal calibration would then decrease by the square root of the number of individual calibrations. Hence, in our case, about 25 calibration profiles of a radiosonde were needed to reduce the insecurity of this seasonal calibration value to ±1%.

## 4. Comparison of GRUAN and Vaisala Product

The radiosonde dataset from global Climate Observing System (GCOS) Reference Upper-Air Network (GRUAN) is considered a high quality radiosonde product. GRUAN data have the advantage of given uncertainties, and humidity measurements are more precise with an additional ground check at 100% relative humidity [30]. However, most of the correction happens at higher altitudes and under conditions that were excluded from the calibration. Thus, a comparison with a calibration using radiosounding measurements as derived by the Vaisala algorithm of RS92 showed very little deviation. In the dark season 2017/2018, the RS41 was used at Ny-Ålesund. The GRUAN product for this type of sonde is not available yet.

Calibrating KARL measurements of the winter season 2016/2017 with both GRUAN-corrected and Vaisala measurements gave a $\widetilde{C_i}$ of 216 g/kg and 214 g/kg in our units, respectively. Thus, the humidity measurements with the uncorrected data showed a slight dry bias. However, even with the biased calibration result, the humidity measurement only deviated by less than 1%. In the year before, a calibration with the Vaisala product yielded a drier result. Differences resulted mainly from the fact that not all profiles were available at GRUAN standard. As we expect the RS41, which was

used in the last dark season (winter 2017/2018), to have further improved water vapor measurements compared to older models, we are confident that results can be derived within an acceptable error margin, even without GRUAN correction.

Figure 4 shows the probability distribution of calibration factors that would be necessary at all heights selected for calibration ($C_i$) for different years and different reference datasets. It is clear that the spread within the respective datasets is much higher than the variation between the different years and different datasets. However, a close inspection of the median values of the lidar calibration constant in Figure 4 reveals that for the winter season 2016/17 we obtained slightly larger calibration value (+2%) than for the other years. For this reason, averaging over longer time periods than one season is not recommended as the sensitivity of the lidar may change. At any change of the system setup, such as the exchange or repair of parts, a separate calibration period should be started.

Comparing the GRUAN and the Vaisala MMR via the lidar, it can be seen in Figure 4 that in the winter season 2015/2016 the Vaisala was about 1% dryer (but still within error bars) while for the season 2016/2017 both datasets were almost identical.

For our best season, 2017/2018 with 21 suitable radiosounding profiles, we were able to determine the median of the seasonal calibration constant to $\pm 1\%$.

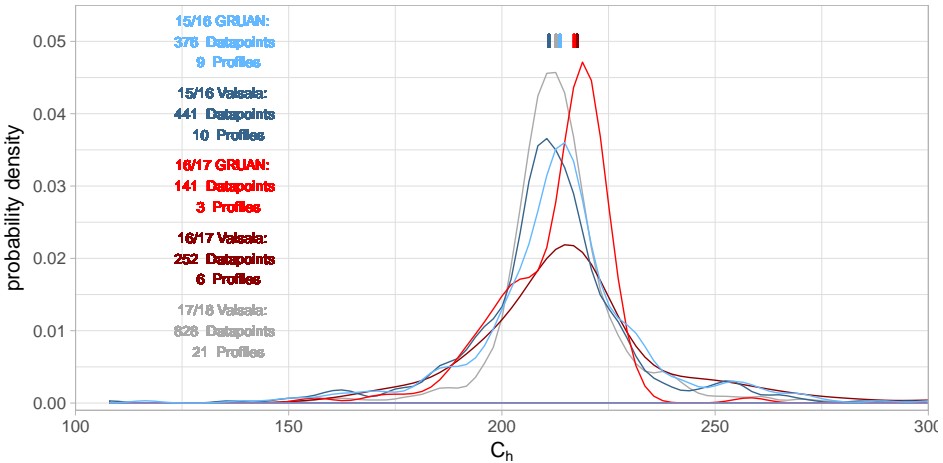

**Figure 4.** Probability density function of $C_i$ for different years and different radiosonde products as reference datasets. Respective median $C$ indicated by an I.

## 5. Treatment of Overlap

In the lowermost meters, the laser beam is not entirely within the telescope field of view. KARL measurements have the disadvantage of different overlap functions in the lowermost 350 m. This may be caused by differential alignment of the 407 and 387 nm channels or by different sensitivities over the surface of the Hamamatsu photomultipliers [32,33]. Effectively, the overlap between the nitrogen and water vapor channel does not cancel out in the lowest altitudes. However, this partial signal still contains information. The shape of the incomplete overlap's signal may change with every setup of KARL. In winter 2017/2018, the ratio between the 387 nm and the 407 nm channel decreased almost linearly with height. Figure 5a shows an exemplary lidar measurement (orange) and the parallel radio sounding measurements (black, dashed) in the lowermost meters. Figure 5b shows the ratio between the measurement pairs for the entire season. There is a clear offset visible with the 407 nm channel being stronger affected by the overlap than the 387 nm channel. A simple linear fit through the ratios at the different altitudes corrects the signal, as shown in green in this figure. The results should be interpreted with caution; however, they do give a good estimation of humidity in the lowermost layers and its variability. The probability density function (PDF) of the ratio of the corrected signal within the incomplete overlap with the radio sounding measurements in this layer (green in Figure 5c) is very similar to the one in the altitudes above (dark green).

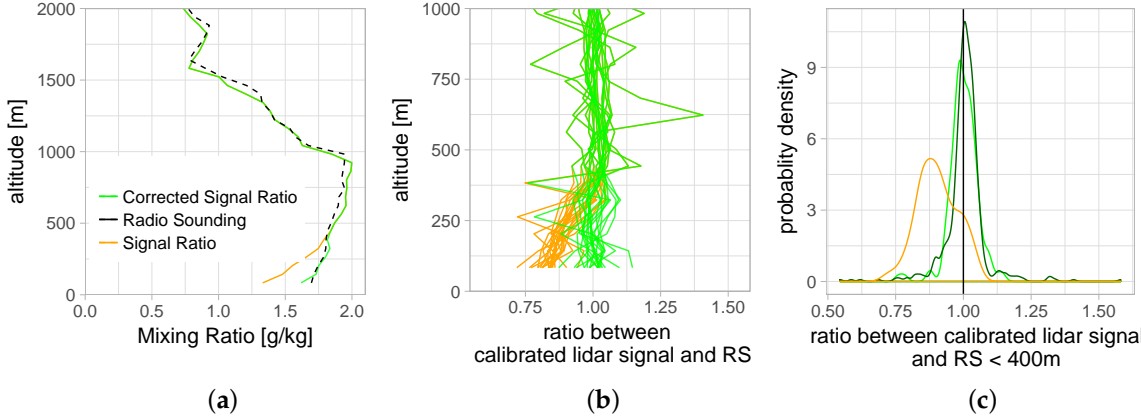

**Figure 5.** (**a**) Exemplary lidar profile used for calibration (20 February 2018); (**b**) ratio between the calibrated lidar signal and radiosounding measurement, with (green) and without (orange) overlap correction for all profiles with parallel measurements in 2018; and (**c**) probability density function (PDF) of ratio between radiosounding and the uncorrected (orange) and corrected (green) lidar signal in the lowermost 400 m. The ratio between the signals in the altitudes chosen for calibration is shown as a reference (dark green).

## 6. Sensitivity Analysis

In this section, an analysis is presented to how a calibration of the lidar water vapor depends on individual error sources. For simplicity, we assumed the radiosonde to be a reference without an inherent error. Basically, the calibration may be distorted by noise in the lidar signal and errors due to the treatment of the Rayleigh attenuation and the aerosol.

First, it should be analyzed what correlation between the mixing ratio from the sonde and the lidar can be expected and how noise in the lidar profile reduces this value. For this purpose, the lidar profile in the dataset with the best correlation to the radiosonde (r = 0.9679) was chosen and the statistic error of the lidar profile ($\Delta P$) defined via the denominator of Equation (4) was calculated. Within 95% (2 $\sigma$) probability, the true lidar profile for all heights z is in the range:

$$P(z) - 2 \cdot \Delta P(z) < P(z) < P(z) + 2 \cdot \Delta P(z) \tag{6}$$

Using a uniform distribution of random numbers between $-2$ and $+2$, 1000 different realizations of the derived noise profile were calculated. An artificial lidar profile, giving a perfect unity correlation to the radiosonde, was modified by those 1000 different realizations of noise and the corresponding correlation to the radiosonde mixing ratio profile was calculated. The result is presented in Figure 6a on the left. It can be seen that our requirement of an SNR > 10 reduced a perfect correlation between sonde and lidar by no more than 3%. For comparison, a probability distribution function (PDF) for SNR > 2 was also calculated. Naturally, a higher noise level would reduce the correlation.

Next, the impact of the Rayleigh scattering was revised. For this, we considered all valid 504 radiosondes from Ny-Ålesund that were launched during 2018. The differences in the ratio of Rayleigh extinction ($\alpha^{Ray}$), i.e. the expression

$$\frac{T_{407}^{Ray}(z)}{T_{387}^{Ray}(z)} = exp\left( \int_{z_0}^{z} (\alpha_{387}^{Ray} - \alpha_{407}^{Ray}) \mathrm{d}\hat{z} \right), \tag{7}$$

was calculated for all profiles up to our upper range of 6 km altitude. The result is plotted in Figure 6b on the right. The impact of different air density profiles on the water vapor was very low. If radiosounding data from the same site as the lidar were available, a possible time difference of several hours between both datasets only contributed to an error in the water vapor retrieval of

around 2‰. Therefore, the insecurity in the MMR profile due to Rayleigh scattering was negligible, if everything else was kept constant.

Finally, the possible impact of the aerosol extinction was analyzed as well. For this, we selected data from March 2018, as March is known to be the month with the highest aerosol pollution in the Arctic due to the phenomenon of Arctic Haze [28,34]. We did so by considering data from the lidar in Figure 6c and from all valid 3425 min of sun-photometer measurements in Ny-Ålesund (Figure 6d) for a broader statistics.

For the lidar, an error of the aerosol extinction coefficient of 100% and an error in the Ångström Exponent of 0.5 were assumed. Which error in the water vapor retrieval occurred from such an error in the extinction was calculated. Figure 6c shows that the median of the error due to extinction was mostly below 0.4% in 6 km and the 95% percentile of the error due to aerosol was below 0.5%. However, it can also be seen in Figure 6c and Equation (3) that an error in the aerosol retrieval did not simply degrade a correlation between the MMR profiles of lidar and radiosonde but, instead, introduced an error that smoothly rises with altitude. (For short distances, the impact of differential extinction was negligible.)

To obtain a realistic upper limit of the impact of aerosol extinction, we also considered sun-photometer data, as this instrument runs on an automated tracking platform continuously and hence provides a broader, more complete dataset. From those measurements, an Ångström exponent was derived. We found a mean $AOD_{500}$ of 0.046 and a mean Ångström exponent of $-1.41$ for March 2018. For all photometer data, the real measured differential AOD between 387 nm and 407 nm was compared to the wrong (artificial) differential AOD that would have been measured by the assumption of Ångström exponent $= -1.2$. This error increased with AOD and the deviation between assumed and real (generally unknown) Ångström exponent. Figure 6d shows how in this worst case estimation, the error due to aerosol extinction was in the order of $\pm 3\%$. However, with our lidar, we do not sound the whole atmosphere, thus the AOD is an upper limit of the aerosol impact.

Figure 6e shows the decrease of a perfect correlation between lidar and sonde due to the combined effects of noise in the lidar data (with SNR > 10) and an assumed 3% error due to the treatment of aerosol. Again, 1000 different realizations of noise and aerosol distributions were considered to produce this plot. It can be seen that a perfect correlation was reduced most likely to a value around 0.975. Pearson correlation coefficients between the logarithm of the MMR profiles of lidar and sonde were below 0.95, hence, cannot be explained by issues in the lidar data or its evaluation. Therefore, the partially very low and rapidly changing correlation coefficients between lidar and sonde in Figure 2 reflected the high variability in the water vapor. Hence, the drift of the sonde and the changing humidity was the most severe error source for calibration. Therefore, the usage of only few co-located radiosondes without checking for identical conditions to calibrate a lidar is generally not sufficient for a satisfying calibration. A comparison between Figure 6a,e reveals that noise in the lidar data was the second most important parameter that influenced the correlation.

Finally, Figure 6f shows the PDF for the precision by which the calibration coefficient between the lidar and the sonde can be determined for our system. The green function refers to the Pearson correlation coefficients of Figure 6e (correlation larger than 0.97). Hence, in our hypothetical best conditions (sonde as an error-free reference, absolutely same air mass, and no instrumental drift of the lidar), it should be possible to constrain the lidar calibration constant to $\pm 2.5$‰ which is the full width at half maximum (FWHM) of the green figure in Figure 6f. The black function refers to realistic cases with a correlation between sonde and lidar >0.95, as in Figure 2. Obviously, this decreased correlation translated into a larger insecurity for the determination of the calibration constant. For this black curve, the FWHM was approximately $\pm 4$‰. Hence, principally an accuracy below 1% could be achieved by averaging of different lidar–sonde profiles if the correlation of each pair is >0.95.

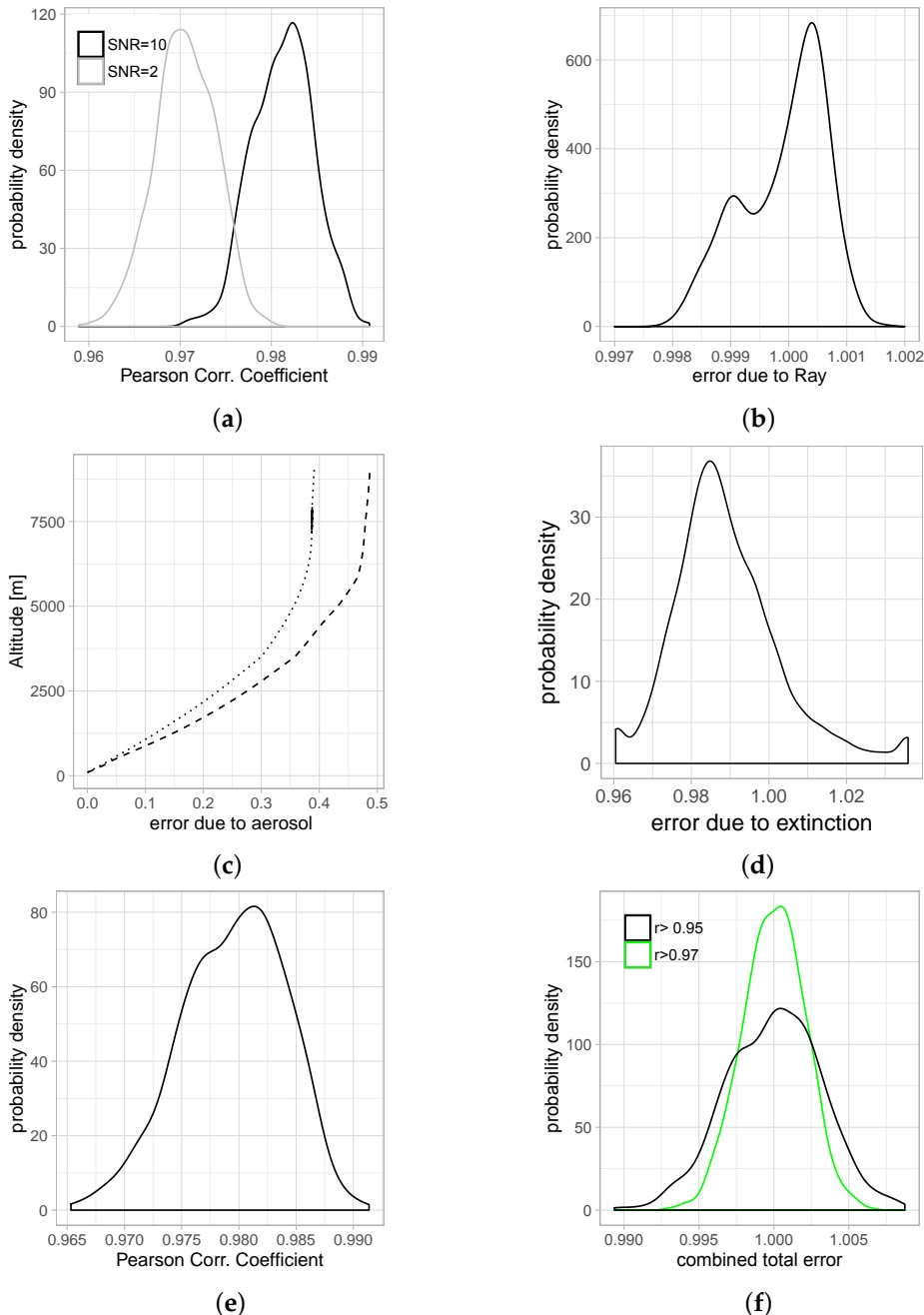

**Figure 6.** (**a**) PDF of Pearson correlation coefficient how different realizations of noise in the lidar degrade a perfect r = 1 correlation; (**b**) PDF of the error due to different Rayleigh atmospheres; (**c**) median (dotted) and 95% percentile (dashed) of the error due to aerosol extinction as a function of altitude; (**d**) PDF of the aerosol induced error from AOD and Ångström exponent; (**e**) PDF of Pearson correlation coefficient of the combined error due to noise and aerosol; and (**f**) accuracy for the determination of the lidar calibration constant.

## 7. Resulting Dataset

The full dataset created by this study can be found in the supplementary materials (doi:10.1594/PANGAEA.896486). Figure 7 shows the resulting water vapor mixing ratio over Ny-Ålesund for three exemplary nights in March 2018. The temporal variability is much higher than it could be resolved by radiosoundings and at a much higher spacial resolution than radiometer measurements. Therefore, a water vapor lidar provides useful and additional information at the site.

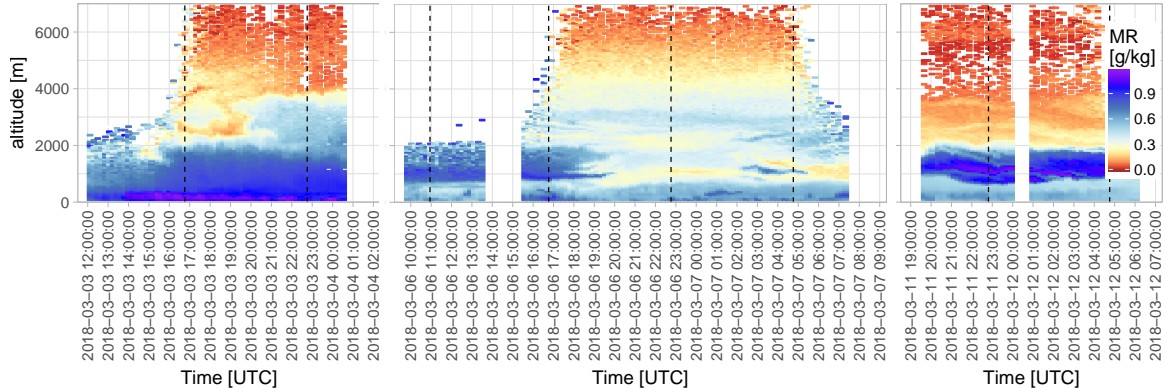

**Figure 7.** Water vapor mixing ratio dynamics over Ny-Ålesund during the nights from 3 March 2018 to 4 March 2018, 6 March 2018 to 7 March 2018 and 11 March 2018 to 12 March 2018. Times of radiosounding measurements indicated by black, dashed line.

Several highly variable moisture inversions are observed. Due to the orography of Spitsbergen in radiosounding profiles, wind shear and elevated temperature inversions can frequently be found up to 2 km altitude [35,36]. However, in Figure 7, it can be seen that moisture inversions and a considerable variability in the MMR also occur in the free troposphere up to about 4 km altitude.

At night-time conditions, MMR profiles up to the polar tropopause in approximately 8 km altitude can be obtained with the given resolution (60 m/10 min). For stratospheric research with a coarser resolution, higher altitudes can be reached.

Since the calibration factor is constant over the season, different measurements can easily be compared to each other; measurements with no simultaneous radiosoundings can be used as well. For this reason, the Raman-lidar can be used for inter-comparisons with other instruments.

## 8. Conclusions

In this work, we revised, assessed and adapted the calibration of lidar water vapor mixing ratio by radiosonde for KARL in a dry, Arctic environment. We applied comprehensible rules on which data points from sonde and lidar are comparable. A correlation of the MMR from the different measurements of over 0.95 was suitable to identify air masses for a valid comparison. Thus, we present a dataset with low and well defined uncertainties that gives additional information to the radiosoundings at high temporal and vertical resolution in the dark period.

With our lidar system being stable over time with mostly non-systematic error sources, averaging several calibrations with independent radiosoundings increased the quality of the calibration. The resulting calibration factor varied by less than ±2% during the seasons investigated in this study. Assessing possible sources of error, we found air mass changes between the compared measurement parts the major source of uncertainty. Uncertainty in aerosol extinction due to an incorrectly assumed Ångström exponent caused an uncertainty in the order of 2%. Potentially inaccurate assumption for Rayleigh scattering could be neglected for this application. The difference between GRUAN and Vaisala radiosounding dataset was below 1% at the ranges used for the calibration.

Considering the error sources discussed in this study, for any other similar calibration problem, we highly recommend examining if the same airmasses are observed and suggest applying similar strategies to avoid these error sources. Furthermore, we suggest averaging over several independent calibrations to decrease the influence of non-systematic error sources.

For the coming years, more measurements with KARL during polar night are scheduled to track humidity in Ny-Ålesund. Especially, with a calibrated water vapor product from the lidar and some knowledge of the physical variability of the humidity at our site, the algorithm of Foth and Pospichal [19] could readily be applied to improve the water vapor retrieval also from a HATPRO radiometer.

**Supplementary Materials:** Datasets created by this study can be found at Kulla, Birte Solveig; Ritter, Christoph (2018): Water vapour mixing ratio over Ny-Ålesund from LiDAR measurements 2015–2018. PANGAEA, https://doi.pangaea.de/10.1594/PANGAEA.896486.

**Author Contributions:** Data curation, B.S.K. and C.R.; Formal analysis, B.S.K.; Funding acquisition, C.R.; Investigation, B.S.K.; Methodology, B.S.K.; Supervision, C.R.; Validation, C.R.; Visualization, B.S.K.; Writing—original draft, B.S.K.; Writing—review & editing, C.R.

**Funding:** This research received no external funding.

**Acknowledgments:** The radiosonde data used for this publication were kindly checked by Siegrid Debatin. The lidar operation was maintained over the years with the help of the station personnel and the engineers Ingo Beninga and Wilfried Ruhe. We gratefully acknowledge the support by the Deutsche Forschungsgemeinschaft (DFG, German Research Foundation)—Projektnummer 268020496—TRR 172, within the Transregional Collaborative Research Center "ArctiC Amplification: Climate Relevant Atmospheric and SurfaCe Processes, and Feedback Mechanisms (AC)3. Two anonymous reviewers supported a quick publication with useful remarks and comments that further improved this paper.

**Conflicts of Interest:** The authors declare no conflict of interest.

## Abbreviations

The following abbreviations are used in this manuscript:

| | |
|---|---|
| A | Ångström exponent |
| $\alpha_{xxx}^{Ray}$ | Rayleigh extinction at the respective wavelength $xxx$ |
| AOD | aerosol optical depth |
| C | calibration constant |
| DJF | December, January, February |
| E | electronic noise |
| FWHM | Full Width at Half Maximum |
| GRUAN | Global Climate Observing System (GCOS) Reference Upper-Air Network |
| KARL | Koldeway Aerosol Raman lidar |
| lidar | light detecting and ranging |
| MMR | mass mixing ratio of water vapor |
| $P_{xxx}$ | lidar signal at the respective wavelength $xxx$ |
| $\Delta P$ | error of the lidar profile |
| PDF | probability density function |
| rh | relative humidity |
| RS | radiosounding, radiosonde |
| $T_{xxx}$ | transmission correction at the respective wavelength $xxx$ |
| $S_{\text{lidar}}$ | signal ratio of transmission-corrected $P_{407}$ to t.-corr. $P_{387}$ |
| SNR | signal to noise ratio |
| $t_i$ | time of measurement closest to RS |
| VMR | volume mixing ratio of water vapor |
| $\rho$ | air density |
| $w$ | water vapor mixing ratio measured by the RS |
| YOPP | Year Of Polar Prediction |
| $z_i$ | a specific height or bin at which criteria are fulfilled |

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
