# Peer review of "Water Vapor Calibration: Using a Raman Lidar and Radiosoundings to Obtain Highly Resolved Water Vapor Profiles"

_remotesensing, doi:10.3390/rs11060616_

Round 1

Reviewer 1 Report

General:

The paper presents Arctic Raman lidar observations in comparison with radiosonde water vapor profiling. The main question after reading the introduction was: What is new? Calibration with radiosondes have been done already 20-30 years ago. Furthermore, the authors do not provide an updated overview of alternative calibration concepts as presented by Foth et al., AMT, 2015, Foth and Pospichal, AMT, 2017, Dai et al, AMT, 2018, and Sakai et al., AMT, 2018. Only after reading Section 2, lines 106 – 124 on page 3-4, the main motivation became obvious. Arctic is a special place, calibration is needed but there are not so many alternatives. And a strong point: The authors have a comparably ‘huge number’ of radiosonde launches and sonde data for comparison, and can thus discuss any technical problem when comparing lidar with radiosonde moisture profiles. These are strong arguments for the paper, … The reader wants to know that: What is the motivation for this paper, what is new, what deserves to be published? …. This must be answered in the Introduction!

The paper is carefully prepared, and also well written.  It is a useful contribution to literature.

Minor revisions are required.

Details:

Simple Summary: The text is rather trivial! Please focus on the special Arctic atmospheric conditions, and that water vapor is an important atmospheric state parameter, and needs to be carefully and continuously measured, … Raman lidar…., high quality ..needs quasi continuous calibration…., we do it with radiosondes… something along these points.

 Abstract: should provide short information about the ‘unique’ goal of the paper (what is the basic motivation for the paper, very short…), the tools (techniques and methods) used , and then the  main findings. Long sentences about the importance of water vapor etc are not appropriate. That can be done  in the Introduction section.

Introduction: Some kind of an exhausting review is needed, and why you selected the radiosonde approach.  Have a look into the papers mentioned above, and also in the references of these papers.

Theory and Background: Keep this section as short as possible. Many parts are already known for a long time. Do we need Eq.(4) and the related discussion?

Calibration:

Table 1… what is rh (please give a clear description)? If rh is relative humidity, I would prefer, 90%, and not 0.9.

Page 5, line 147. How is S_lidar and w expliciteyl defined?

Page 7, line 183: Is  that  possible to reduce the uncertainty in the Raman lidar calibration by means of radiosonde comparison to 1=%? To my opinion there is no way to be better than 3-5% because of permanently slight changes in the performance of the Raman lidar. 

Treatment of overlap

Page 8, line 226: .. a simple linear fit through the ratios… … corrects the signal.

I have my doubt that this is a good solution. Why should that properly work? If the overlap profile is different for the two signal channels, then there is no hope for a good correction for overlap values of 0.8 and smaller (in the near range of the lidar).

Sensitivity analysis:

Pages 9-11: Please state as often as possible … What is new compared to other radiosonde calibration papers? There are many points to my opinion. It is necessary to avoid the potential  impression: They ‘copy and paste’ the Raman lidar calibration method already used since 30 years to Arctic observations and that’s it.

Resulting Dataset:

This section is quite short, and partly contra productive to the foregoing argumentation. The message should be: Intercomparison is always useful and required to keep the quality of the Raman lidar observations at a high level…..although the calibration factor is quite constant over the season… so that…..

Author Response

Dear Sir or Madame, 

Thanks a lot for Your very helpful review.

In the attached Word document you will find our reply to your comments.

Sincerely 

Christoph Ritter & Birte Kulla

Reviewer 2 Report

The authors in this study are presenting the calibration of a water vapour channel of a Raman lidar operating in the polar Arctic, against radiosoundings. Although relevant studies already exist in the scientific literature, this manuscript is mostly focused on the various uncertainties that may introduced under the calibration process. The manuscript is well written but in order to be improved I would kindly suggest the following:

line 7: "mistakes" -> "sources of error" 

line 8: "best year":  Please define the term "best year". I have the felling that this is a typo. Consider correcting it or be more specific.

line 20: "best season" : Please define the term "best season". Maybe the authors meant to say "best case"?

line 44: "to measuring" -> "to measure"

line 70: "color" -> "wavelength"

line 72: "recording" -> either "collecting" or "detecting"

line 77: "bin" -> "height bin"

Eq. 4: is the product c E a constant value through your measurement time period? In that case your could define here what is the value of this parameter. 

line 122: "agree well enough". I am missing why these 21 simultaneous profiles agree well enough. Can you please specify in which sense they agree specifically? Maybe the authors meant that they fulfill specific requirements?

Table 1: How realistic is this time period of 1 hour? For the moment it looks a little bit arbitrary the 1 hour time scale. The authors are kindly requested to explain this further. I have also the same comment for the first requirement namely SNR of P407>10. The value of 10 looks again arbitrary. 

line 144: "....allowing a maximum... ". It is not so clear to me how this maximum height requirement set by the authors is eliminating the possibility of radiosonde drift? To my knowledge radiosondes may be also equipped with GPS allowing the estimation of their horizontal distance from the launching site. If such a dataset is available to the authors it will be helpful for them to use it. 

line 147: "...water vapour mixing ratio...": Please specify here also that this quantity is retrieved from the radiosondes

 line 166: "... do not show a good ....". Any possible explanation on this? Such a case seems to be the one during 20180-03-07 05:00 UTC. The 1/8 balloon failure looks like a high rate to me. 

Figure 3: In this figure both black and grey lines are referred to as seasonal median values. It is not so clear to me which is their difference. Additionally the text in the manuscript describing this figure, do not make this clear either. I would kindly suggest to the authors to rephrase the lines 173-183, since this figure is important.

line 186: Provide the full name of the acronym GRUAN, when this is used for the first time (i.e line 63).

line 194: Are there any units? Please specify them.

lines 218-219:   

Indeed this is the general situation: The distance of full overlap depends both on the geometry of the emission-detection module (laser-telescope) but also on the characteristics of the wavelength separation unit (e.g. mounting precision of the optics, maximum acceptance angle of the interference filter, temperature of operation etc.). Moreover depends also on the inhomogeneity of the photomultiplier. 

The authors are kindly suggested to include the following references here: 

Kokkalis P.,  Using paraxial approximation to describe the optical setup of a typical EARLINET lidar system, Atmos. Meas. Tech., 10, 3103–3115, 2017, https://doi.org/10.5194/amt-10-3103-2017, 

and 

Simeonov, V., Larcheveque, G., Quaglia, P., van den Bergh, H., and Calpini, B.: Influence of the photomultiplier tube spatial uniformity on lidar signals, Appl. Opt., 38, 5186–5190, https://doi.org/10.1364/AO.38.005186, 1999.

line 219-220: The authors are kindly suggested to provide information (along with the appropriate reference) relevant to their approach for correcting the lidar signals for full overlap. This is not found in the manuscript but in figure 5 the authors are demonstrating the improvement on the retrievals at lowest atmospheric heights, after overlap correction.

line 236: Eq. 4 defines the SNR of the system and not the error of the lidar signal. Please be more specific on your statement.  Moreover, to which error the authors are referring to (Statistical, Systematic) ?

line 237: If I understood well, the authors here are describing their approach based on a monte carlo simulation. Why the performed only 100 different runs ? Is this an adequate number for leading to  solid and trustful results ? 

Figure 6a: Why the maximum value of Pearson correlation in this figure is 0.990 and not 1?

 Eq. 7: Consider re writing the formula of subtracting the extinctions inside a parenthesis. This subtraction is integrated. 

Figure 6b: Please specify on the x axis of this figure if this is absolute error, percent error ?

line 247: Consider mentioning if for each sensitivity run you kept the rest of the requirements constant or not.  

line 259: Is this error related to the overlap of the lidar instrument?

line 278: Correct to the appropriate symbol "greater than".

Figure 6 legend: "perfekt" -> "perfect" 

lines 285-287: Consider providing a reference for this statement

Author Response

Dear Sir or Madame, 

Attached You find our replies to Your comments. 

Thanks a lot for Your fast and helpful review.

Sincerely

Christoph Ritter & Birte Kulla

Round 2

Reviewer 2 Report

The authors have sufficiently addressed my concerns in their new manuscript version.Therefore, in my opinion, the revised manuscript can be now accepted for publication in remote sensing journal of MDPI.